# Analysis of Resource Potential of Emergent Aquatic Vegetation in the Curonian Lagoon of the Baltic Sea

**Yuliya Kulikova** [1,*] **, Julia Gorbunova** [2] **, Sergey Aleksandrov** [2,3] **, Marina Krasnovskih** [4] **, Valentin Gurchenko** [1] **and Olga Babich** [1]

1 Institute of Living Systems, Immanuel Kant BFU, 236016 Kaliningrad, Russia; benzvalen@gmail.com (V.G.); olich.43@mail.ru (O.B.)

2 Shirshov Institute of Oceanology, Russian Academy of Sciences, Nakhimovsky Prospect, 36, 117997 Moscow, Russia; julia_gorbunova@mail.ru (J.G.); hydrobio@mail.ru (S.A.)

3 Atlantic Branch of VNIRO (AtlantNIRO), 236022 Kaliningrad, Russia

4 Department of Inorganic Chemistry, Chemical Technology and Technosphere Safety of Perm State National Research University, 614990 Perm, Russia; krasnovskih@yandex.ru

* Correspondence: kulikova.pnipu@gmail.com; Tel.: +7-9127849858

**Abstract:** This paper presents results of an aquatic vegetation resource potential assessment. The study is aimed at assessing the perspective of biotechnological approaches to reducing the biogenic pollution of water bodies by the removal of aquatic vegetation. The article analyzes the dominant species of aquatic vegetation in the Curonian Lagoon, and their productivity and resource potential. It was established that the concentrations of protein and fat in the biomass of four dominant plants are extremely low, making it impossible to speak of their values in terms of biomass processing. Based on elemental composition, we can conclude that the biomass of *Phragmites australis* should have a high calorie content because it has a high carbon and hydrogen content (49.6% and 7.1%, respectively), resulting in a high energy value. Synchronous thermal analysis revealed that the maximum energy values of biomasses of *Phragmites australis* and *Scirpus lacustris* have a net calorific value of 12.62 and 12.55 MJ/kg, respectively. At the same time, the biomass of these plants has a low ash content (around 6.6–7.6%) and a low sulfur content (less than 0.41%). An analysis of the composition of aquatic vegetation biomass samples allowed us to establish that, given the permissible collection time (no earlier than September), the search for directions in the utilization of aquatic vegetation should focus on processing cellulose with the production of crystalline cellulose, biochar, or biofuel via anaerobic digestion. The removal of the excess biomass of aquatic vegetation (*Phragmites australis*) will allow the reduction of the nitrogen and phosphorus load in the water body by 140 kg/ha and 14 kg/ha, respectively.

**Keywords:** industrial environmental aquatic chemistry; emergent aquatic vegetation; water pollution; resource potential; biomass





## 1. Introduction

The Curonian Lagoon is the largest lagoon in the Baltic Sea (area 1584 km$^2$, volume 6.2 km$^3$), with predominantly freshwater conditions due to poor water exchange with the sea and high river discharge [1,2]. Significant pollution with biogenic elements (mainly with waters of the Neman River), as well as the specific hydrological conditions, led to the accumulation of nutrients (biogenic elements) and severe eutrophication of the lagoon. The Curonian Lagoon belongs is one of the most highly eutrophic water bodies (hypertrophic type) based on its concentration of nutrients, primary production, and phytoplankton abundance [3]. The combination of high concentrations of biogenic elements in the water and bottom sediments, freshwater conditions, poor water exchange, and the intense summer heat create the conditions for the annual "blooming" of cyanobacteria [4–6].

The coastal zone plays a significant role in the functioning of the Curonian Lagoon ecosystem. Shallow water, low salinity, and high nutrient content create favorable conditions for the development of emergent aquatic vegetation, which, according to modern data, includes up to 225 species from 142 genera and 56 families. The Curonian Lagoon is dominated by common reed (*Phragmites australis* (*Cav.*) *Trin. Ex Steud.*), a key species in coastal and wetland ecosystems that forms reedbeds. A significant share of the Russian coast of theCuronian Lagoon (from the Neman River delta to the southern part of the Curonian Spit) is heavily covered by common reed and lakeshore bulrush (*Scirpus lacustris* L.). The emergent aquatic plants grow in belts, forming biocenoses.

Emergent aquatic vegetation is an important component of the lagoon ecosystem, forming part of the overall biological productivity of the Curonian Lagoon. The quantitative characteristics of emergent aquatic vegetation for the Baltic region are scarce, with most data focusing on common reeds (average 500–1000 $g/m^2$) [7,8]. According to preliminary estimation, common reed and lakeshore bulrush form 4.9 and 1.4 thousand tons of air-dry mass per year, respectively, in the Russian part of the Curonian Lagoon [2], which later can be consumed by animals and included in the detrital food chain after bacterial decomposition.

The reproduction of fish populations is highly dependent on emergent aquatic vegetation. The spawning and feeding of juvenile and adult fish of the most important commercial species (roach, bream, and others) occurs in its thickets. Thickets of emergent aquatic vegetation also play an important environmental role in the formation of bird and mammal biological diversity as 23 species of birds nest in them [9]. A significant part of the Baltic Sea region's coastal zone is protected [10].

Emergent aquatic vegetation influences the physical and chemical properties of the lagoon's water. It can retain organic and mineral suspensions as a result of mechanical filtration and accumulate various chemical compounds (nutrients, heavy metals, pesticides, oil products, etc.), that improve water quality [11].

Common reed is a highly productive plant with a yield of 3 to 30 t/(ha·yr) [12] and is used for a variety of purposes such as building material, fuel, animal feed, and raw material for pulp production. Traditionally, reed is used for roofing: from one hectare of reed up to 100 $m^2$ of roof can be covered [8,13].

The main disadvantage of using reed as fuel is its low density. In this regard, it is only recommended to use unprocessed reed for heating in areas no further than 50 km from where it grows [13]. Reed has an energy content that is comparable to wood when it is grounded and compressed into briquettes or pellets, achieving a high calorific value of 17–18 MJ/kg [8,13].

The cellulose content of reed is 33–59%, making it suitable for use as a raw material in the pulp and paper industry. However, reed pulp and paper are not currently produced in Europe due to a lack of sufficient raw materials, as well as for environmental reasons, but similar enterprises continue to operate actively in China, using up to 2.7 million tons of reed per year [13].

These traditional uses of reed have declined significantly nowadays, but reed has emerged as a promising resource for energy production, such as biogas and bioethanol, as well as nutrient removal from water supplies and wastewater treatment [8].

Reed has advantages as a renewable energy source when compared to first-generation biofuel crops. First, this energy crop does not require the use of agricultural land. The second concerns the conservation of carbon stocks in peatlands, which, when drained, release large amounts of greenhouse gases as a result of the decomposition of peat [13].

In this vein, the main purpose of this work is to analyze the dominant species of aquatic vegetation in the Curonian Lagoon, and their productivity and resource potential. The results of this work can be used for the development of biotechnological approaches in the biogenic pollution management of water bodies by the timely removal of aquatic vegetation.

## 2. Materials and Methods

### 2.1. Plant Sampling Procedure

The emergent aquatic vegetation of the Curonian Lagoon was studied using generally recognized manuals [14,15]. During the growing season (May–November 2022), samples of different species were collected in the model plot on a monthly basis. To calculate the biomass, vegetation was cut from a 0.25 m² area several times, then the wet and air-dry mass were calculated.

The model plot (54°56′00″ N, 20°41′45″ E) was located on the southern coast of the Curonian Lagoon in the area of the Primorskaya harbor from its outer and inner sides (Figure 1). The bay's soils are predominantly silty, sandy on the outside, and shelly in some places. The model plot was chosen in such a way that the thickets of emergent aquatic vegetation were well developed.

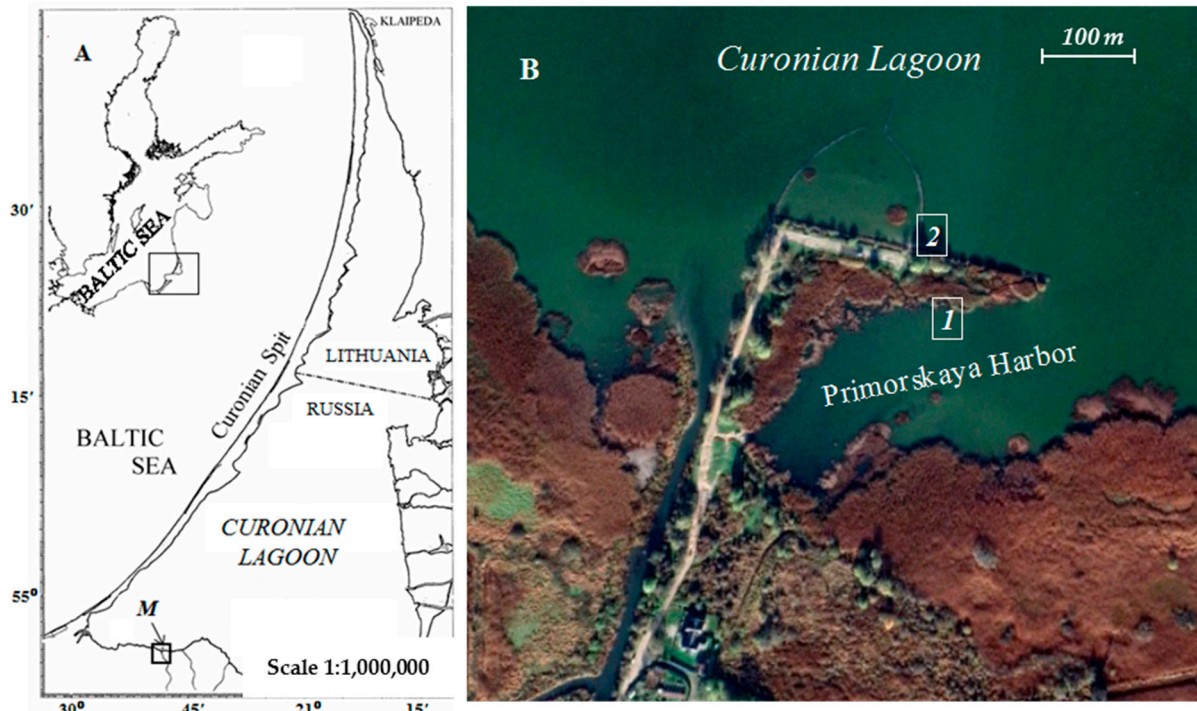

**Figure 1.** Location of the model plot (M) in the Curonian Lagoon (**A**) and sampling scheme in Primorskaya Harbor (**B**) (1, 2—sampling plots) (Google Earth image, 11 June 2022).

### 2.2. Determination of the Physicochemical Composition of Aquatic Plants

The samples were dried at 60 °C to and mechanically grounded into a homogeneous mixture of particles. The protein content was determined using the Kjeldahl method [16]. Mineralization of the sample was carried out at the temperature of 360 °C for 180 min. The sample weight was 0.5 g. Using an automatic Kjeldahl apparatus, ammonia was distilled into boric acid.

Fat content was determined gravimetrically. Fat extraction was performed with diethyl ether [17].

The content of reducing carbohydrates was determined using picric acid [18]. The sample weight was 0.5 g. Samples were first hydrolyzed in 10 mL of a 10% sulfuric acid solution for two hours. The pH of the hydrolyzate solution was adjusted to neutral. At the end of the hydrolysis, the probe was filtered, and the volume was adjusted to 50 mL. One milliliter of the hydrolyzate solution was used for the reaction with picric acid. The optical density calibration was obtained by measuring standard glucose solutions.

The ash content was determined gravimetrically. Ashing was performed in porcelain crucibles in a muffle furnace at 600 °C. The average ash values were calculated based on the results of two parallel determinations.

The elemental analysis of plant biomass was performed using the elemental analyzer Vario EL Cube (Elementar Analysensysteme GmbH, Langenselbold, Germany) CHNS based on the area of the chromatographic peaks of $N_2$, $CO_2$, $H_2O$, and $SO_2$.

Calorific value and thermal properties of biomass were estimated by NETZSCH STA 449C Jupiter (NETZSCH-Gerätebau GmbH, Selb, Germany) based on synchronous thermal analysis. Tests were made in an oxidizing (air) environment.

## 3. Results and Discussion

### 3.1. Results of Emergent Aquatic Vegetation Species Diversity and Biomass Assessment

Emergent aquatic vegetation were studied in the southern part of the Curonian Lagoon. This area is characterized by a relatively shallow depth (less than 4 m), freshwater conditions, and a long period of water renewal due to its remoteness from the Klaipeda Strait and the mouth of the Neman River. The concentrations of nutrients (nitrogen, phosphorus) in the southern region as a whole were the lowest across the Russian water area on average from 2019 to 2022, especially in spring, due to the lack of a significant inflow of river discharge [19]. However, nitrogen and phosphorus concentrations may have been higher near the shore due to small river inflow, particularly that of the Malaya Moryanka River, which flows into the model plot.

The study of emergent aquatic vegetation was conducted from the outer and inner sides of the Primorskaya harbor, which are separated by an L-shaped pier and have different environmental conditions. The inner side of the harbor, which had more stable hydrodynamic conditions, had the highest species diversity and vegetation biomass. The belt of emergent aquatic vegetation was 5 to 30 m wide (Figure 2a).

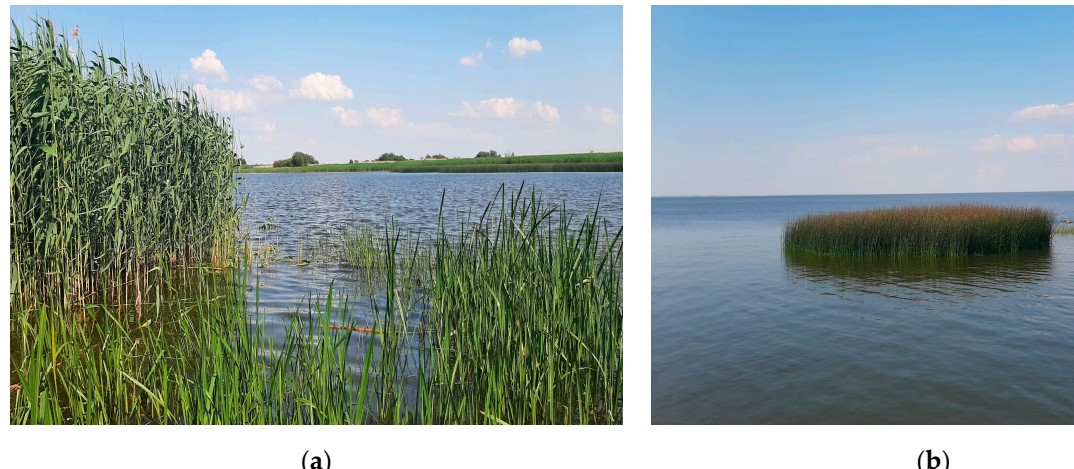

(**a**)　　　　　　　　　　　　　　　　　　　　　(**b**)

**Figure 2.** Thickets of emergent aquatic vegetation in the Curonian Lagoon: a continuous belt in the harbor of Primorskaya (**a**), separate clumps on the outside (**b**) during the period of intensive growth in June 2022.

From the shore to depths of 0.8–1.0 m the details of the growths of the following species were established: *Phragmites australis*, *Scirpus lacustris* L. and *Typha angustifolia* L., *Sparganium erectum* L. and *Butomus umbellatus* L. Aquatic plants were also represented by *Nuphar luteum* L., *Potamogeton perfoliatus* L., and *Ceratophyllum demersum* L. There was no continuous belt of aquatic vegetation from the outer side of the harbor because of regularly wave activity in the Curonian Lagoon. *Scirpus lacustris* L. and *Butomus umbellatus* L. clumps with diameters of 10–15 m were located at a distance of 20–50 m from the shore (Figure 2b).

The quantitative development of various species of emergent aquatic vegetation varied significantly (Table 1). Common reed had the highest biomass (965 g/m$^2$), being the most

dominant and abundant species in the coastal zones of the Baltic Sea [8]. Narrow-leaved cattail also had a high share of biomass in this plot, which exceeded that of lakeshore bulrush (367 g/m$^2$). The simplestem bur-reed was also observed to have a high biomass (223 g/m$^2$). The biomass of floating and submerged aquatic plants (yellow water-lily, pondweed) was much lower (15–75 g/m$^2$).

**Table 1.** Biomass of the main species of emergent aquatic vegetation in 2022 (Curonian Lagoon, Primorskaya Harbor).

| Species | Air-Dry Biomass, g/m$^2$ | |
|---|---|---|
| | September | Average (May–November) |
| *Phragmites australis (Cav.) Trin.ex Stend.* | 1433 | 965 |
| *Scirpus lacustris* L. | 672 | 367 |
| *Typha angustifolia* L. | 1386 | 712 |
| *Sparganium erectum* L. | 408 | 223 |
| *Butomus umbellatus* L. | 85 | 102 |
| *Nuphar luteum* (L.) *Smith* | - | 75 |
| *Potamogeton perfoliatus* L. | - | 35 |
| *Ceratophyllum demersum* L. | - | 15 |

The values obtained in the model plot are similar to those previously obtained for the Russian part of the Curonian Lagoon (common reed 993 g/m$^2$, lakeshore bulrush 470 g/m$^2$), as well as results of common reed biomass assessments for the coasts of Finland, Sweden, and Estonia (average 500–1000 g/m$^2$) [7,8].

Seasonally, the amount of biomass increased from the start of the growing season in Spring to the end of the hydrological Summer in September (Figure 3). In August–September, the common reed biomass reached 1433 g/m$^2$, while narrow-leaved cattail and lakeshore bulrush biomass reached 1386 and 672 g/m$^2$, respectively. Vegetative growth ceased in October–November due to external environmental factors, and biomass degradation began (Figure 4). The time of degradation onset and its intensity varied significantly for different species. Lakeshore bulrush, simplestem bur-reed, and flowering rush reduced biomass by 50% by November, and narrow-leaved cattail by 30%. At the same time, common reed biomass changed insignificantly, only by 3% during the Summer period. The thickets of this plant were well preserved for a long time after the end of active vegetation, making this species very convenient for winter collection. A slight change in reed biomass was reported during the winter period (March) compared to the summer maximum (July), which allowed it to be used in economic activities [7].

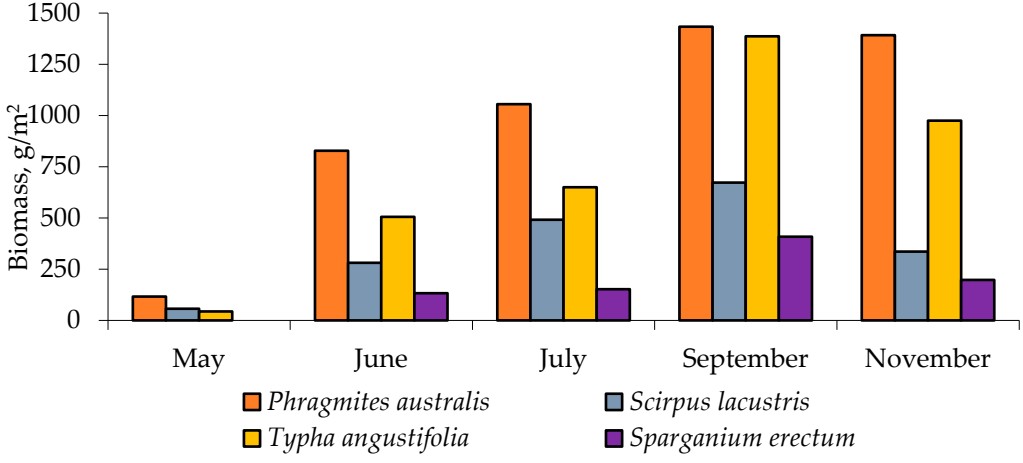

**Figure 3.** Seasonal dynamics of the biomass of the main species of emergent aquatic vegetation.

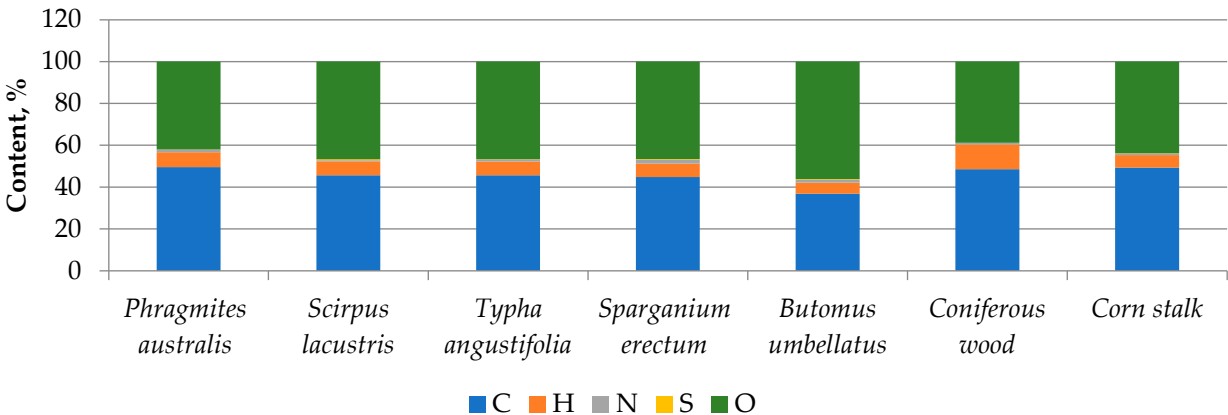

**Figure 4.** The content of macro elements in the biomass of the studied species of aquatic vegetation (in % ash-free dry mass).

Reed has potential as a renewable resource due to its high growth rate, stable annual harvesting, and growth in coastal areas that are not used for high-value crops. According to preliminary estimates, the area of common reed thickets in the Russian part of the Curonian Lagoon is 480 ha, and lakeshore bulrush is 305 ha, representing a reserve of 4.9 and 1.4 thousand tons of air-dry mass, respectively [2].

Studies have shown that the largest biomass of emergent aquatic vegetation is observed in late Summer–early Autumn (August–September). Reed harvested in summer has a higher nutrient content than its dry biomass in winter. It can be used as feed for farm animals, for the production of fertilizers or compost, as well as biogas [13,20]; in particular, such recommendations were given by experts for lagoonal and estuarine ecosystems of the South Baltic [21]. Reed collected in the Kaliningrad region can potentially be used for these purposes. It should also be noted that during the summer, other mass species (lakeshore bulrush, narrow-leaved cattail, simplestem bur-reed, etc.) can be collected, the biomass of which degrades in late autumn.

Despite the positive aspects of the collection of emergent aquatic vegetation in the summer, a number of important limitations exist for the coastal zone which must be considered. Spawning and feeding of fry and fish of the most important commercial species (roach, bream, and others) occurs in thickets of emergent aquatic vegetation. Summer collection may have a negative impact on the reproduction of fish in the Curonian Lagoon, which is an important fishery reservoir in the Baltic region. Thickets of emergent aquatic vegetation also play an important role for nesting.

The negative impact of collecting emergent aquatic vegetation on ichthyofauna, birds, and other animals can be reduced if the collection is done during the winter [7,8]. However, it should be taken into account that in the winter period (October–May) in the coastal zone of the Curonian Lagoon, only common reed can be collected, while the surface biomass of other mass species, such as lakeshore bulrush, decomposes by winter. Reed thickets tolerate winter pruning well due to their well-developed underground rhizome system, which can even increase aboveground biomass in the following growing season, according to some studies [13]. The winter collection is used in most areas of the Baltic Sea, minimizing conflict with nature conservation. In the coastal zone of the Curonian Lagoon, the use of emergent aquatic vegetation can be important for solving urgent problems such as improving the biological diversity of coastal areas, reducing eutrophication, and improving water quality, in addition to solving economic problems.

According to a number of data, the yield of the biomass of aquatic plants can be significantly affected by a number of factors, but to a greater extent, the level of salinity, the degree of water body pollution, the direction and intensity of coastal territory use, the occurrence of fires, and the mowing of vegetation.

Livestock grazing in the coastal zone, and trampling or mechanical damage by boats to coastal vegetation undoubtedly leads to a change in productivity in this area; however, at the moment, there are no prerequisites for a significant change in the level and direction of economic use in the coastal territories of the Curonian Lagoon. The salinity of the water in the Curonian Lagoon, according to long-term observation (an over-twenty-year period), is a fairly stable factor [22,23] and varies within 0.5–1.9 ppt with an average salinity of 1.2 ppt. According to the data of Huang et al. [23] and Lissner et al. [24], salinity change up to 5 ppm does not have a significant effect on the development of the common reed biomass. Thus, a change in the productivity of aquatic plant biomass in the Curonian Lagoon as a result of salinity impact is not expected in the near future. The occurrence of fires is a spontaneous and difficult-to-model process. Its influence on the dynamics of aquatic plant biomass development is simply not taken into account, even in most models.

The emergent aquatic vegetation of the Curonian Lagoon is represented by perennial herbaceous species with rhizomes, which are well known for a fairly stable species composition and biomass production, even in the long-term in a certain area.

The study period in 2022 in the Kaliningrad region was characterized by fairly typical weather conditions, which made it possible to consider this year as a model year for the preliminary assessment of the abundance and chemical composition of aquatic vegetation in the Curonian Lagoon. Analysis of satellite data (Google Earth images) did not show significant changes in the distribution of plant communities in this area over a long period. The absence of the expected significant change in environmental and economic aspects that could have a significant impact on the biomass of aquatic plants, as well as the relative stability of their biocenosis, allows us to draw the following conclusion: additional studies will only provide additional clarifying data, but will not give new contradictory conclusions.

Thickets of emergent aquatic vegetation can perform an important reducing and eutrophicating function for water bodies since, due to their large biomass and high growth rate, they effectively remove excess nutrients. In Summer, the nutrient content of reed averages 1% nitrogen and 0.1% phosphorus [7,21]. The peak of nutrient accumulation in the upper part of the reed occurs in July or August; later on, nutrients accumulate in the roots. To reduce eutrophication and improve water quality, nutrients must be removed through the collection of aboveground biomasses from emergent aquatic vegetation. The biomass of overground common reed obtained on the model plot (1433 g/m$^2$ or 14 t/ha) contained up to 140 kg/ha of nitrogen and 14 kg/ha of phosphorus, which can be removed from the body of water. According to the optimistic scenario, reed harvesting in the Baltic region would reduce nitrogen concentrations by 1% and phosphorus concentrations by 3% annually, contributing to meeting the target levels set by the HELCOM Baltic Sea Action Plan [25]. The effect of regular reed collection can be much greater in a closed shallow lagoon, such as the Curonian Lagoon. Some important advantages of using reed thickets to remove excess nitrogen and phosphorus from the Curonian Lagoon are the low capital and operating costs, especially considering the further use of cut reed biomass as a promising renewable resource in the region's economy.

### 3.2. Assessment of the Physicochemical Properties of Emergent Aquatic Vegetation

An assessment of the physicochemical properties of aquatic vegetation was obtained from samples taken in September, since this month seems to be the most optimal for sampling emergent aquatic vegetation, if it is used for the production of biogas, biofuel, or cellulose. The results of the analysis of the biogenic elements and biochemical compositions in the biomasses of different species of aquatic vegetation in the Curonian Lagoon are presented in Table 2 and Figure 5.

Based on the biogenic elements content, we can conclude that *Phragmites australis* biomass should have a high caloric content because it contains a large amount of carbon and hydrogen (49.56% and 7.09%, respectively), resulting in a high energy value (even higher than that of coniferous wood). The high content of nitrogen and sulfur in the biomass of *Sparganium erectum*, *Butomus umbellatus*, and *Phragmites australis* (1.78%, 1.42, and 1.25%,

respectively) correlates well with the high content of proteins found in the biomasses of these species of aquatic vegetation (6.25%, 5.20, and 5.85%, respectively) (Tables 2 and 3).

**Table 2.** Biogenic elements composition in the biomasses of different species of aquatic vegetation in the Curonian Lagoon.

| Sample | Source | Content, % | | | | |
|---|---|---|---|---|---|---|
| | | C | H | N | S | O (calc.) |
| *Phragmites australis* | Experimental results | 49.56 ± 0.98 | 7.09 ± 0.08 | 1.25 ± 0.57 | 0.08 ± 0.01 | 42.01 |
| *Scirpus lacustris* | | 45.65 ± 0.93 | 6.67 ± 0.12 | 0.55 ± 0.03 | 0.41 ± 0.17 | 46.73 |
| *Typha angustifolia* | | 45.66 ± 3.13 | 6.55 ± 0.28 | 0.95 ± 0.10 | 0.37 ± 0.06 | 46.47 |
| *Sparganium erectum* | | 44.79 ± 0.53 | 6.48 ± 0.11 | 1.78 ± 0.25 | 0.32 ± 0.20 | 46.63 |
| *Butomus umbellatus* | | 36.82 ± 0.87 | 5.28 ± 0.09 | 1.42 ± 0.11 | 0.27 ± 0.05 | 56.21 |
| Coniferous wood | [26] | 48.56 | 11.84 | 0.7 | 0.06 | 38.84 |
| Corn stalk | [27] | 49.30 | 6.00 | 0.70 | 0.11 | 43.89 |

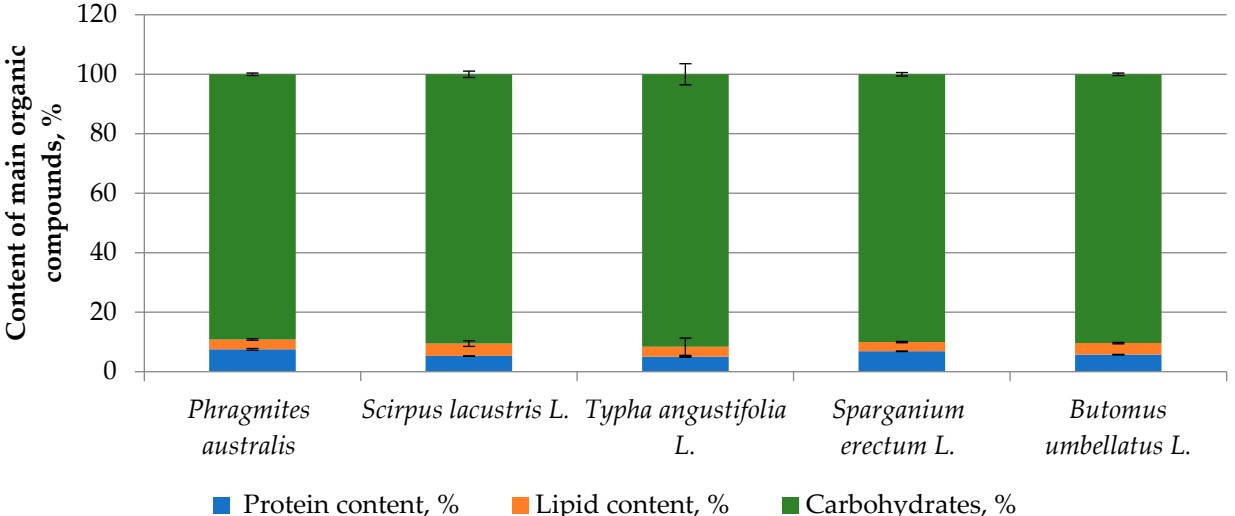

**Figure 5.** The content of macro elements in the biomasses of different species of aquatic vegetation (in % ash-free dry mass).

**Table 3.** Biochemical composition in the biomasses of different species of aquatic vegetation in the Curonian Lagoon.

| Sample | Protein Content | Lipid Content | Carbohydrates | Ash Content, d.m.% |
|---|---|---|---|---|
| | Ash-Free Dry Mass, % | | | |
| *Phragmites australis* | 5.85 ± 0.29 | 3.03 ± 0.28 | 81.30 | 8.83 ± 0.04 |
| *Scirpus lacustris* | 4.84 ± 0.01 | 3.81 ± 0.95 | 82.69 | 8.66 ± 0.07 |
| *Typha angustifolia* | 4.65 ± 0.16 | 3.1 ± 0.29 | 84.51 | 7.75 ± 0.48 |
| *Sparganium erectum* | 6.25 ± 0.09 | 2.81 ± 0.21 | 81.92 | 9.02 ± 0.28 |
| *Butomus umbellatus* | 5.20 ± 0.08 | 3.47 ± 0.26 | 81.48 | 9.85 ± 0.07 |

If we compare the content of proteins and fats in the biomass of *Phragmites australis*, we can see that Beyzi et al. [28] obtained almost the same fat content and ash content of 2.98 and 9.45%, respectively, but the protein content was almost three times higher (17.44%). The high content of protein in the biomass of *Phragmites australis* discovered by Beyzi et al. [28] was due to the fact that the biomass was sampled in June before the flowering of plants, compared to September in our work, when the biomass had already begun to wither.

A comparison of the physicochemical composition of the *Typha angustifolia* biomass with results obtained by other scientists [29] allows us to discuss the similarity of the results;

the ash content obtained by the authors is at the level of 7.33–9.4%, while the cellulose content is 66–89%.

The high amount of protein found in the *Sparganium erectum* biomass was expected as, in the literature, there is a high protein content for all species of the genus *Sparganium* at a level of 7.6–13.2% [30].

Reliable data in open sources to compare the physicochemical composition of *Butomus umbellatus* and *Scirpus lacustris* with the available data were not found.

Based on the composition of the studied aquatic vegetation biomasses, we can conclude that the search for ways of utilizing biomass, taking into account the permissible collection time (no earlier than September), should focus on processing cellulose with the production of crystalline cellulose, biochar, or biofuel through anaerobic digestion. The protein and fat concentrations in the biomass are extremely low, making it impossible to consider it valuable in terms of obtaining these components. *Sparganium erectum* L. and *Phragmites australis* are of the highest nutritional value, since these plants have a high content of protein and lipids.

When assessing the resource potential of a plant biomass, its calorific value and thermal properties were also evaluated. The results are presented in Figure 6 and Table 4. The number of main peaks on the differential scanning calorimetry curve was used to calculate the number of main stages of biomass destruction, and weakly expressed stages were excluded from the analysis. Table 4 contains the temperatures at the start and end of the stage ($t_1$ and $t_2$), as well as the temperature at which the maximum rate of sample weight loss ($t_{max}$) was observed. The total loss of the sample at this stage was indicated to assess the completeness of the process ($\Delta m$).

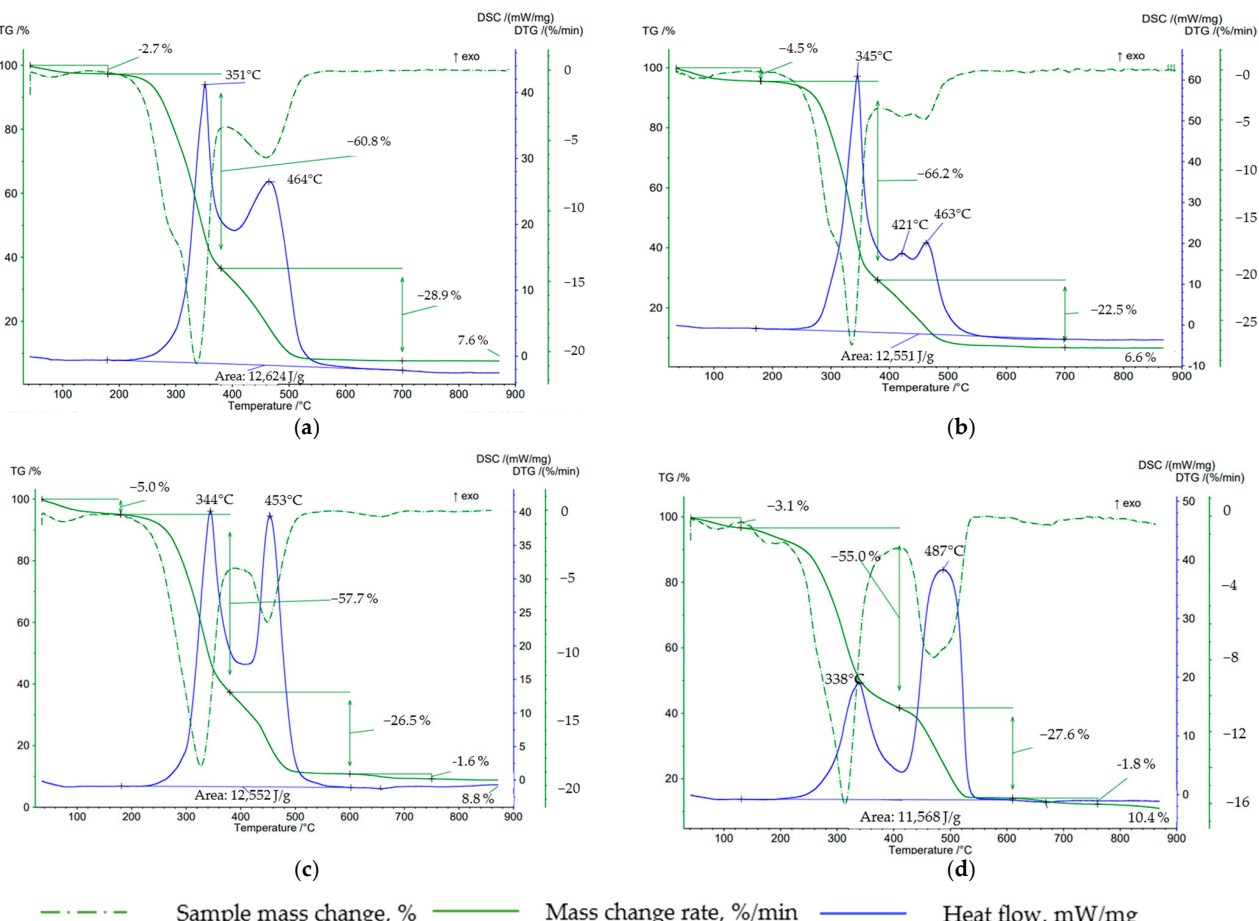

**Figure 6.** Results of synchronous thermal analysis of plant biomass in air: (**a**) *Phragmites australis*; (**b**) *Scirpus lacustris*; (**c**) *Typha angustifolia*; and (**d**) *Sparganium erectum*.

**Table 4.** Results of the thermal analysis of plant samples in the air.

| Sample | Number of Main Stages | $t_1$ | $t_2$ | tmax | Δm, % | Rest, % | NCV, KJ/g |
|---|---|---|---|---|---|---|---|
| *Phragmites australis* | 1 | 220 | 380 | 351 | 60.8 | 7.6 | 12.62 |
| | 2 | 380 | 700 | 464 | 28.9 | | |
| *Scirpus lacustris* | 1 | 250 | 380 | 345 | 66.2 | 6.6 | 12.55 |
| | 2 | 380 | 450 | 421 | 22.5 | | |
| | 3 | 450 | 560 | 463 | | | |
| *Typha angustifolia* | 1 | 210 | 380 | 344 | 57.7 | 8.8 | 12.55 |
| | 2 | 380 | 590 | 453 | 28.5 | | |
| *Sparganium erectum* | 1 | 240 | 410 | 338 | 56.7 | 10.4 | 11.57 |
| | 2 | 410 | 540 | 487 | 24.6 | | |
| *Butomus umbellatus* | 1 | 130 | 370 | 323 | 42.8 | 19.1 | 10.31 |
| | 2 | 370 | 510 | 468 | 26.4 | | |

Four out of the five aquatic plants biomass samples showed two-stage degradation under oxygen conditions, and only *Scirpus lacustris* had a three-stage decomposition.

The first stage for all samples occurs in the temperature range of 180–380 °C, accompanied by an average weight loss of 60% compared to the initial sample. At this stage, weak polar bonds are broken with the formation of predominantly hydrogen-containing compounds ($H_2O$, $CH_4$). The first peak should be associated with the destruction of the bulk organic compounds, including cellulose, xylan, and lignin [31].

The second stage occurs in the temperature range of 380–520 °C and is accompanied by an average weight loss of 29%. At this stage, the decomposition of the carbon bonds occurs with the breaking of C-C bonds and the formation of $CO_2$ [32].

Both stages are exothermic, as can be seen from the two distinct peaks in the thermal effects curve. It is clear that the biomass with the minimum ash content and minimum biomass decomposition temperature has the best thermal properties. According to these parameters, the *Phragmites australis* and *Scirpus lacustris* biomasses have the best properties, having an ash content of 7.6% and 6.6%, respectively, and a temperature of maximum weight loss of 351 °C and 345 °C, respectively. It is also significant that when burning *Sparganium erectum*, we observe the maximum rate of weight loss, and, accordingly, the maximum heat flux only at a temperature of 487 °C, and the ash content for this plant reaches 10.4%, i.e., this type of biomass is not suitable for direct combustion.

The presence of the third additional peak during the destruction of *Scirpus lacustris* is most likely associated with the destruction of sulfate, which typically remains unchanged up to a temperature of 427–513 °C and only at these temperatures begins to turn into sulfides [33].

Based on the results of the synchronous thermal analysis, it can be concluded that biomasses of *Phragmites australis* plants have the most optimal thermal properties. The net calorific value (NCV) of these biomasses is 12.62 MJ/kg, which is comparable to the calorific value of *Abies alba* wood, which is estimated at 18.9 MJ/kg [34]. A temperature of destruction in the range of 344–351 °C should be considered optimal because, at this temperature, the maximum destruction rate is achieved.

**4. Conclusions**

The Curonian Lagoon is the largest lagoon in the Baltic Sea. Extensive shallow water, fresh water, and a high content of nutrients create favorable conditions for the development of emergent aquatic vegetation. The coastal zone of the Curonian Lagoon is dominated by the common reed (*Phragmites australis*), producing a biomass above 1000 g/m$^2$. Lakeshore bulrush (*Scirpus lacustris*), narrow-leaved cattail (*Typha angustifolia*), and simplestem bur-reed (*Sparganium erectum*) also have a high biomass among emergent aquatic vegetation, the biomass of which in some areas in Summer can reach 400–1000 g/m$^2$. The largest biomass of emergent aquatic vegetation is observed in late Summer–early Autumn (August–

September). Eutrophication of the Curonian Lagoon waters is characterized by a maximum level and an annual "bloom" of cyanobacteria, which has a negative impact on the entire aquatic ecosystem. Thickets of emergent aquatic vegetation can perform important functions in the reduction of eutrophication since, due to their large biomass and high growth rate, they effectively remove nutrients

The conducted assessment of aquatic plants' composition allowed us to establish that the nutritional use of excess plant biomass is not advisable due to their low nutrient content (especially taking into account the safe harvest period of no earlier than September). A more promising direction is cellulose processing with the production of crystalline cellulose, biochar, or biofuel via anaerobic digestion.

Based on the analysis of biomass productivity and elemental composition, it was found that the collection of even only *Phragmites australis* (the dominant species) will ensure the annual removal of 14 kg of phosphorus and 140 kg of nitrogen per hectare.

In addition to economic goals, the harvesting and subsequent use of common reed, which forms thickets in the coastal zone, can be important for improving coastal biodiversity, reducing eutrophication, and improving water quality.

A study of the composition and yield of biomasses of aquatic plants in the Russian part of the Curonian Lagoon was carried out for the first time and its main purpose was to assess the prospects of using dominant species to obtain valuable products. Obviously, the data of one year need to be verified and confirmed by the results of long-standing research. However, if we rely on the experience of other scientists, the composition of grass vegetation biomass has not changed significantly from year to year. Pociene and Kadziuliene [35] studied the content of cellulose, lignin and hemicellulose in reed canary grass and tall fescue, which was grown as feedstock or for combustion, for two years. It was found that the variability of cellulose, lignin and hemicellulose content in the biomasses collected in the same months, but in different years, did not exceed 5%, which is within the error of the determination methods. In addition, studies by a number of authors have suggested that the yield of aquatic plant biomass in the absence of extreme factors remains fairly stable from year to year [21,36]. Taking into account the fact that the data obtained will be used to determine the general directions of use of the biomasses of the studied aquatic plants, fluctuations within 10–15% of the content of individual components will not affect the accuracy of the conclusions.

**Author Contributions:** Conceptualization, O.B. and S.A.; methodology, J.G.; investigation, J.G., Y.K., V.G. and M.K.; resources, O.B.; writing—original draft preparation, J.G., S.A. and Y.K.; writing—review and editing, O.B.; visualization, J.G., S.A. and Y.K.; project administration, O.B.; funding acquisition, O.B. All authors have read and agreed to the published version of the manuscript.

**Funding:** The study was carried out with the financial support of the Ministry of Science and Higher Education of the Russian Federation, project number FZWM-2023–0003.

**Institutional Review Board Statement:** Not applicable.

**Informed Consent Statement:** Not applicable.

**Data Availability Statement:** Not applicable.

**Conflicts of Interest:** The authors declare no conflict of interest.

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
