# Peer review of "Analysis of Resource Potential of Emergent Aquatic Vegetation in the Curonian Lagoon of the Baltic Sea"

_water, doi:10.3390/w15112136_

Round 1

Reviewer 1 Report

This manuscript presents “Analysis of the Resource Potential of Curonian Lagoon Emergent Aquatic Vegetation”.

The methods seem to be sound. I found the manuscript properly written and interesting however some points need to be amended. 

-          species names used in charts should be in italics

-          sometimes there is a lack of proper discussion, it should be extended

-          adding the references site would improve quality

-          biotechnological use is poorly marked but the results do not allow it, they are done reliably but their scope is too narrow to determine such potential

Other comments:

-          The purpose of this study is clear and well-addressed in the introduction.  

-          It is difficult to be innovative in such research and the manuscript contains quite a narrow spectrum of analysis however it meets all the standards of this type of research. There will always be gaps in this kind of research, there is still much to be done. 

-          Many publications only touch on one aspect of resource potential research and here we have a few ones. I think that such studies fill such a gap well, but are of local interest.

I recommend accepting this manuscript after minor corrections.

Author Response

Dear reviewer Many thanks for the great work done.
Your comments allowed us to significantly improve our article.
A more detailed response to comments on points is provided in the attachment.
Best regards

Reviewer 2 Report

  1. This works analysis and discusses the Curonian Lagoon emergent aquatic vegetation. Its findings are a good reference for the related states to develop their ecosystem, environment, and wastewater treatment.  
  2.  It can also help the reader understand the dominant species of aquatic vegetation and develop biotechnological approaches to biogenic pollution management.

Author Response

Dear reviewer.

Thank you very much for your review and appreciation of our paper. 

Best regards

Kulikova Yuliya

Reviewer 3 Report

Dear Editor!

 The paper "Analysis of the Resource Potential of Curonian Lagoon Emergent Aquatic Vegetation" presented by Yuliya Kulikova et al. in the journal "Water" is an interesting multidisciplinary study of the dominant species of aquatic vegetation in the Curonian Lagoon, their productivity and resource potential. The authors have done extensive work on the analysis of the composition of aquatic vegetation biomass samples.

However, when reviewing the work carefully, the authors should make the following points to clarify the essential points:

1) The authors have concentrated their work on only one year of study, which makes it difficult to transfer findings and conclusions about bigger periods and perspectives.

2) How can the situation of only one year of data analysis be clarified in order to establish a framework for this type of study? The authors do not write about this, which makes such a selective study impractical and unpromising.

3) What environmental, economic or other factors might influence this one-year study data? The authors have not done this clarification, which makes the data set of the limited period of the study very poor, selective and unsuitable for prediction.

4) The authors cite predominantly old literature sources.

This work requires substantial revision.

Author Response

Dear Reviewer

Thank you for your attention to our article. All your comments have been taken
into account and the text of the article has been amended based on your recommendations. More detailed responses to comments are provided in the
attached file.
Best regrads.

Reviewer 4 Report

A brief summary

The manuscript studies the analysis of the resource potential of Curonian Lagoon aquatic plants. The topic is interesting and little known in aquatic science. The writing is clear. The aim of article should be more clearly defined. The aim is very general - should be specified. Also, logical consistency between aims and results-discussion sections should be improved. New aspects should be emphasized. However, some revision will be necessary to improve the readability of the manuscript. I recommend a major revisions of the manuscript and re-write and re-organization before it will be acceptable for publication. I have few specific comments, which might improve the manuscript.

Material and methods

The methods are not described accurately enough. There is no information about the number of plots/plants samples.

Specific comment

Line 25: NCV - the acronym requires explanation

Line 81: HCV – again, acronym requires explanation

Figure 1 - the map has no legend and scale

Figure 6 – the fonts are too small and therefore unreadable

Table 3 – ash free dry mass; Line 271 – ash-free mass; The Authors use two different terms, please calrify.

Author Response

(The authors gave the same response as above.)

Round 2

Reviewer 3 Report

Dear Editor!

The authors did not correct or improve the text of the presentation of the research material in accordance with the comments of the reviewer. I cannot agree with this.

Author Response

Dear Reviewer We tried to answer your comments more fully and correct the article as much as possible.
Please find reply in the attached file.
Sincerely, Yuliya Kulikova
